# MyLynch: A Patient-Facing Clinical Decision Support Tool for Genetically-Guided Personalized Medicine in Lynch Syndrome

**DOI:** 10.3390/cancers15020391

**Published:** 2023-01-06

**Authors:** Stephen T. Knapp, Anna Revette, Meghan Underhill-Blazey, Jill E. Stopfer, Chinedu I. Ukaegbu, Cole Poulin, Madison Parenteau, Sapna Syngal, Eunchan Bae, Timothy Bickmore, Heather Hampel, Gregory E. Idos, Giovanni Parmigiani, Matthew B. Yurgelun, Danielle Braun

**Affiliations:** 1Dana-Farber Cancer Institute, 450 Brookline Ave, Boston, MA 02215, USA; 2University of Rochester School of Nursing, 601 Elmwood Avenue, Rochester, NY 14642, USA; 3University of Pennsylvania, Philadelphia, PA 19104, USA; 4Northeastern University, 360 Huntington Ave, Boston, MA 02115, USA; 5City of Hope, 1500 E Duarte Rd, Duarte, CA 91010, USA; 6Harvard T.H. Chan School of Public Health, Harvard University, 677 Huntington Ave, Boston, MA 02115, USA

**Keywords:** Lynch Syndrome (LS), intervention, cancer penetrance, cancer risk, genetically-guided personalized medicine (GPM), clinical decision support (CDS) tool, risk communication, cascade testing, germline genetic testing, genetic counseling

## Abstract

**Simple Summary:**

Lynch syndrome (LS) is associated with varying cancer risks depending on which of the five causative genes harbors a pathogenic variant; however, lifestyle and medical interventions provide options to lower those risks. We developed MyLynch, a patient-facing clinical decision support (CDS) web application that applies genetically-guided personalized medicine (GPM) for individuals with LS. MyLynch informs LS patients of their personal cancer risks, educates patients on relevant interventions, and provides patients with adjusted risk estimates depending on the interventions they choose to pursue. MyLynch can improve risk communication between patients and providers while also encouraging communication among relatives with the goal of increasing cascade testing. As genetic panel testing becomes more widely available, GPM will play an increasingly important role in patient care, and CDS tools offer patients and providers tailored information to inform decision-making. MyLynch provides personalized cancer risk estimates and interventions to lower these risks for patients with LS.

**Abstract:**

Lynch syndrome (LS) is a hereditary cancer susceptibility condition associated with varying cancer risks depending on which of the five causative genes harbors a pathogenic variant; however, lifestyle and medical interventions provide options to lower those risks. We developed MyLynch, a patient-facing clinical decision support (CDS) web application that applies genetically-guided personalized medicine (GPM) for individuals with LS. The tool was developed in R Shiny through a patient-focused iterative design process. The knowledge base used to estimate patient-specific risk leveraged a rigorously curated literature review. MyLynch informs LS patients of their personal cancer risks, educates patients on relevant interventions, and provides patients with adjusted risk estimates, depending on the interventions they choose to pursue. MyLynch can improve risk communication between patients and providers while also encouraging communication among relatives with the goal of increasing cascade testing. As genetic panel testing becomes more widely available, GPM will play an increasingly important role in patient care, and CDS tools offer patients and providers tailored information to inform decision-making. MyLynch provides personalized cancer risk estimates and interventions to lower these risks for patients with LS.

## 1. Introduction

Lynch syndrome (LS), formerly called hereditary non-polyposis colorectal cancer (HNPCC), affects approximately 1 out of 279 people and is associated with significantly increased risks, and potential earlier onsets, of a number of cancers, including colorectal (CRC), endometrial, ovarian, gastric, and more [1,2,3]. It is a hereditary disease with an autosomal dominant pattern caused by a pathogenic variant (PV) (primarily missense, nonsense, frameshift, and splice site change variants) in one of four mismatch repair (MMR) genes: *MLH1*, *MSH2*, *MSH6*, and *PMS2*, or a PV with a large deletion in the 3’ region of the *EPCAM* gene which, in turn, causes epigenetic silencing of MSH2 [2,4,5].

Evidence regarding the types of cancer associated with PVs in the various LS susceptibility genes, and the magnitude of cancer risks associated with these PVs, is emerging at a fast rate. However, even for the five LS genes, the number of gene-cancer combinations to consider is large, information is dispersed over a vast number of articles, the quality of the available literature is uneven, and the data presented are seldom directly applicable to precision prevention decisions, which require absolute risk [6]. Thus, patients can not always take maximum advantage of the impressive scientific advancements in this field.

A counseling session for a newly diagnosed or potential LS patient often includes risk estimates and a discussion regarding specialized screening and risk-reducing interventions available for the LS-associated cancers [7]. Genetics providers are well trained in communicating risk to patients; however, this task remains challenging for several reasons, including the impact emotions have on medical decisions; the degree of relevance and personalization of the information, as perceived by the patient; varying numerical and graphical literacy; and social factors [8,9,10,11,12]. Risk visualizations have been shown to be highly effective at enhancing this type of challenging communication [9,10]. Possible risk-reducing interventions may include regularly scheduled colonoscopies at increased frequencies and starting at younger ages than the general population, an aspirin regimen, weight loss, and/or prophylactic surgeries [13,14,15,16,17].

A patient’s blood or saliva sample can be used to identify the presence of germline PVs on single genes, multiple gene panels, exomes, or whole genomes using microarrays, sequencing, and karyotyping [18]. For individuals who test positive for LS, cascade testing of at-risk relatives is recommended [19,20]. This cost-effective strategy provides great benefits to untested family members; however, it relies on the ability and level of comfort the patient has with disclosing their results to all at-risk family members [20].

Cancer risk estimates and preventative actions can be individually tailored because the level of associated cancer risk varies widely depending on which gene harbors the PVs, as well as the patient’s sex, age, and other characteristics [1,7,21]. According to Dominguez-Valentin (2020), a female LS patient with a *MLH1* PV is estimated to have a 37% cumulative risk of endometrial cancer by age 75, whereas if they instead carried an *EPCAM* PV, then their risk would only be around 13%. The same work also reports the cumulative risk of CRC for a male LS patient with a *MLH1* PV by the time they are 50 years old to be around 34% but, for females with a *MLH1* PV, it is only around 21% by age 50 [1]. In genetics, a penetrance is an effective measure of cancer risk. A penetrance is the proportion of individuals with a certain genotype who express a specific phenotype. In this context, a genotype can be defined by an LS causative gene with a PV and each cancer type is a different phenotype. Tools that provide cancer risk based on a patient’s genetic and personal information fall under the umbrella of genetically-guided personalized medicine (GPM) [6]. GPM clinical decision support (CDS) tools provide customized health care to patients while addressing resource constraints on providers and the lead-lag effect that is common between research and common medical practice [6,22,23].

The ‘All Syndromes Known to Man Evaluator’ (ASK2ME) tool (https://ask2me.org/) (accessed on 2 December 2022) is a GPM CDS tool that provides cancer risk estimates for carriers of germline PVs; however, it was designed to be clinician-facing and contains highly technical language [24]. The Prospective Lynch Syndrome Database (PLSD) also has a tool that provides similar risk information specifically for people with LS in a simple interface (https://sigven78.shinyapps.io/plsd_v4/) (accessed on 2 December 2022). Although this tool is useful for some users, more context would be needed for many patients to grasp the content easily, and information is lacking for *EPCAM* carriers [1]. Neither of these tools includes the ability to modify individual risks by incorporating the effects of risk-reducing interventions. The goal of this work was to develop a CDS web application (web app) that applies GPM to benefit LS patients through a patient-focused iterative design process that utilized focus groups and cognitive qualitative interviews. We named our web app MyLynch (https://MyLynch.org) (accessed on 2 December 2022), and its specific aims were to:1.Inform patients of their cancer risks, beyond just those cancers most classically associated with LS, using state-of-the-art risk prediction models in an easily understood format tailored for each patient and accessible to them in their home or at their clinician’s office [1].2.Educate patients on lifestyle and medical interventions available to them and the magnitude of risk reduction these interventions can have on their cancer risks [13,14,15,16,17].3.Improve risk communication between clinicians and patients while addressing resource limitations on medical providers and potential gaps between clinicians’ personal knowledge of gene-cancer associations and the rapidly developing and cutting-edge body of research on cancer genetics [6].4.Aid in communication among relatives with the goal of increasing cascade testing and improving health outcomes among at-risk relatives [19].

In the Materials and Methods section, we will describe the literature review and statistical techniques used to estimate personalized cancer risks, with and without risk-reducing interventions, as well as the design of the patient focus groups and cognitive interviews used to develop the web app. In the Results section, we will summarize the database of cancer penetrances, which incorporated the five LS genes and nine different cancer types that were used to estimate personalized future cancer risks. We will also describe the risk-reducing intervention options included in MyLynch: colonoscopies, aspirin, weight loss, and prophylactic oophorectomies and hysterectomies. We will review the patients’ responses and feedback to prototype versions of MyLynch, how those responses drove improvements and the development of new features, and describe the resulting app with screen shots and visualizations of personalized risk estimates. In the Discussion and Conclusions sections, we will review how we accomplished the four aims outlined in the previous paragraph, how our tool effectively communicates patient risks through interactive visualizations, and limitations of the tool.

MyLynch provides LS patients with personalized cancer risks based on their unique profiles and educates patients on available interventions and the extent to which those interventions can lower their cancer risks. MyLynch was well received by the patients in our focus groups and interviews and has the potential to improve communication between patients and clinicians, as well as encourage patients and their family members to adopt risk-reducing strategies and undergo cascade testing.

## 2. Materials and Methods

Having established the aims of the MyLynch web app, the next steps included: (1) identify studies that provide cancer risk estimates by at least age, sex, and gene for each of the five LS genes, and, where possible, incorporate risks stratified by race and ethnicity, then apply statistical methodology to convert these risks into age-specific conditional penetrance (ACP) estimates; (2) conduct a literature review of available risk-reducing interventions, which can be used to modify the penetrance estimates established in step 1; (3) develop the statistical methodology and write a program “back-end”, which can customize absolute risk estimates to individual users based on their gene, sex, current age, race, ethnicity, surgical history, and cancer history, and which modifies those absolute risk estimates based on impacts of the interventions identified in step 2; and (4) create a user interface (UI or “front-end”) for the back-end designed specifically for LS patients using an iterative development process that leverages patient focus groups and interviews with additional input from the expert clinical team members. The focus groups and one-on-one cognitive qualitative interviews of LS patients were approved by the Institutional Review Board at the Dana–Farber Cancer Institute.

### 2.1. Cancer Penetrance

To tailor clinical decisions for an individual to support LS management and cancer prevention efforts, our goal was to estimate personalized absolute probabilities of developing various LS-associated cancer types between an individual’s current age and some future age, based on that individual’s genotype, sex, race, and ethnicity. Next we provide a detailed definition of these probabilities, which requires defining the notation of Table 1.

In the context of this work, a penetrance is the probability of an individual with a PV in one of the LS genes, *j*, developing a specific type of cancer, *c*. An age-specific penetrance provides the probability of an individual developing cancer at some discrete age. An age-specific penetrance can be personalized based on factors such as genotype $G_{j}$, sex *S*, race *R*, and ethnicity *E* by conditioning on these factors, which makes it an age-specific conditional penetrance (ACP). Following the notation of Table 1, $P(T_{c}=a|G_{j}=g,\;S=s,\;R=r,\;E=e)$ is the probability of developing cancer *c* at age *a* conditional on genotype *g* for gene *j*, sex *s*, race *r*, and ethnicity *e*. We refer to this probability as the ACP.

For an individual who is cancer-free through their current age, ACPs can be used to evaluate future risk, that is, the probability of developing a cancer *c* between the individual’s current age $a_{cur}$ and some future age $a_{fut}$ for an individual with genotype *g* for gene *j*, sex *s*, race *r*, and ethnicity *e* [24,25]. This is given by:(1)$$\begin{array}{c} P(a_{cur}<T_{c}\leq a_{fut}|G_{j}=g,S=s,R=r,E=e,T_{c}>a_{cur})=\\ \;\;\;\;\;\;\;\;\;\;\;\;\;\;\frac{\sum_{a=a_{cur+1}}^{a=a_{fut}}P\left(T_{c}=a|G_{j}=g,S=s,R=r,E=e\right)}{S_{c,g,s,r,e}\left(a_{cur}\right)} \end{array}$$

The denominator of Equation (Equation 1), the survival function from cancer *c*, can also be expressed in terms of ACPs as follows [24,25]: (2)$$S_{c,g,s,r,e}\left(a_{cur}\right)=1-\sum_{a=1}^{a_{cur}}P\left(T_{c}=a|G_{j}=g,S=s,R=r,E=e\right)$$

We estimated ACPs based on a rigorous literature review of LS gene-cancer associations using PubMed and Embase keyword searches for terms related to LS, genetics, and cancer [24]. We identified studies that provided cancer risk estimates for cancers that had a significantly higher risk of occurrence in individuals with LS, as compared to individuals without LS, and where those cancer risks were stratified by at least age, sex, and gene.

Highlighting the differences in risk between non-LS individuals and those with LS became an important mechanism for communicating risk in MyLynch. Therefore, we also acquired ACP estimates for the general population for each cancer identified in the literature review, and these penetrance estimates were derived from the Surveillance, Epidemiology, and End Results Program (SEER) database [25]. Due to the rarity of the PVs that cause LS, we assumed that the number of LS patients in the SEER database is sufficiently small, and therefore, we assumed that the SEER penetrance estimates for the general population are equivalent to the penetrance estimates for individuals without LS [24]. The cumulative risks for individuals without LS can also be estimated using Equation (Equation 1) by setting *g* to 0 for any gene *j*.

The cancer risk estimates derived from SEER were stratified by race and ethnicity, which enabled us to condition the ACPs by race and ethnicity for the general population. For LS cancer studies that reported cancer risks relative to the general population in the form of relative risks (RRs), odds ratios (ORs), or hazard ratios (HRs), we were able to derive ACPs conditional on race and ethnicity [24,25,26].

The ACPs extracted from the literature sources for LS patients and the ACPs from SEER for the general population formed the basis of the MyLynch database [26]. We refer to these ACP estimates in the database as baseline ACPs to reflect that these values do not factor in the effects of interventions.

### 2.2. Intervention Effect Estimates

A literature review for modifiable risk factors (body mass index [BMI]) and proven risk-reducing behaviors (aspirin use, colonoscopic screening, and prophylactic hysterectomies and salpingo-oophorectomies) was systematically conducted using PubMed and Embase searches, and these searches targeted studies of LS patients and the general population. Studies and their results were screened for relevance to LS patients, sample size, and significance. Each study identified reported either a RR, OR, or HR. We developed a statistical methodology to apply these risk modifiers to the baseline ACPs to obtain post-intervention ACPs. The following equations were developed for this methodology to estimate risk for one cancer, one gene, and one intervention that can be generalized to any cancer type, any of the five LS genes, and any intervention.

*For studies reporting a RR,* a baseline ACP value can be multiplied by the reported RR to estimate the post-intervention ACP, as represented in Equation (Equation 3).
(3)$$\begin{array}{c} P(T_{c}=a|G_{j}=g,S=s,R=r,E=e,\Phi_{k}=1)=\\ \;\;\;\;\;\;\;\;\;\;RR_{c,a,s,r,e,g,k}\times P\left(T=a|G_{j}=g,S=s,R=r,E=e,\Phi_{k}=0\right) \end{array}$$

The left-hand side of Equation (Equation 3) is the probability of developing cancer *c* at age *a* conditional on genotype *g* for gene *j*, sex *s*, race *r*, ethnicity *e*, and application of intervention *k*.

*For studies reporting an OR,* where the intervention *k* has a binary application, meaning it is either applied or not (e.g., the patient either receives regular colonoscopies or not) then $RR_{c,a,g,s,r,e,k}$ needs to be estimated from the reported OR using Equation (Equation 4) before utilizing Equation (Equation 3) [27].
(4)$$\begin{array}{c} RR_{c,a,g,s,r,e,k}=OR_{c,a,g,s,r,e,k}/[\left(1-P\left(T_{c}=a|G_{j}=g,S=s,R=r,E=e,\Phi_{k}=0\right)\right)+\\ \left(OR_{c,a,g,s,r,e,k}\times P\left(T_{c}=a|G_{j}=g,S=s,R=r,E=e,\Phi_{k}=0\right))\right] \end{array}$$

For interventions that are not binary, such as change in body weight, we first used Equation (Equation 5) to calculate $OR_{c,a,g,s,r,e,k}$ as follows [16]:(5)$$OR_{c,a,g,s,r,e,k}={\left(1+\sigma_{v_{k}}OR_{c,a,g,s,r,e,k,v_{k}}\right)}^{|\Delta v_{k}|}$$

*For studies reporting an HR,* where the intervention *k* has a binary application, then the baseline ACPs in the database first must be converted to hazard rates (λ) using Equation (Equation 6) then multiplied by the reported hazard ratio (HR) using Equation (Equation 7) and, finally, converted back into ACPs using Equation (Equation 8).
(6)$$\lambda_{c,a,g,s,r,e,0}=\frac{P(T_{c}=a|G_{j}=g,S=s,R=r,E=e,\Phi_{k}=0)}{1-\sum_{t=1}^{a-1}P\left(T_{c}=t|G_{j}=g,S=s,R=r,E=e,\Phi_{k}=0\right)}$$
(7)$$\lambda_{c,a,g,s,r,e,1}=HR_{c,a,g,s,r,e,k}\times \lambda_{c,a,g,s,r,e,0}$$
(8)$$\begin{array}{c} P\left(T_{c}=a|S=s,G_{j}=g,R=r,E=e,\Phi_{k}=1\right)=\lambda_{c,a,g,s,r,e,1}\times \left(1-\sum_{t=1}^{a-1}\lambda_{c,t,g,s,r,e,1}\right) \end{array}$$

To make Equation (Equation 7) applicable for reported HRs applied on a continuous scale, such as change in BMI, we first used Equation (Equation 9) to calculate $HR_{c,a,g,s,r,e,k}$ [15].
(9)$$HR_{c,a,g,s,r,e,k}={\left(1+\sigma_{v_{k}}HR_{c,a,g,s,r,e,k,v_{k}}\right)}^{|\Delta v_{k}|}$$

*Adjusted baseline ACPs.* Studies used to estimate the CRC baseline ACPs did not control for colonoscopies or adherence to a regular colonoscopy schedule; therefore, we had to assume that the study samples consisted of a mix of patients who adhered to a regular colonoscopy schedule and those who did not [4,28,29,30]. Furthermore, because the RR for regular colonoscopies compares patients who do and do not follow a regular colonoscopy schedule, we cannot use Equation (Equation 3) to directly apply this RR value to the baseline ACPs [13]. To address this issue, we estimated the proportion of patients that participated in the gene-cancer studies on CRC who did not follow regular colonoscopies by using other literature sources [31,32]. By solving a system of equations involving Equations (Equation 3) and (Equation 10) below, we were able to estimate adjusted baseline ACPs of CRC for individuals who do not follow a regular colonoscopy schedule:(10)$$\begin{array}{l} P(T_{c}=a|G_{j}=g,S=s,R=r,E=e)=\\ \;\;\;\;\left(1-p_{c,a,g,s,r,e,k}\right)\times P\left(T_{c}=a|G_{j}=g,S=s,R=r,E=e,\Phi_{k}=0\right)+\\ \;\;\;\;\;\;\;\;\;p_{c,a,g,s,r,e,k}\times P\left(T_{c}=a|G_{j}=g,S=s,R=r,E=e,\Phi_{k}=1\right) \end{array}$$

The left-hand side of Equation (Equation 10) is a baseline ACP determined from a gene-cancer study on CRC that had a mix of patients who implemented intervention *k* and those who did not (e.g., a mix of individuals who adhered to and did not adhere to a regular colonoscopy schedule).

CRC and endometrial cancer risks vary due to changes in BMI and weight; therefore, the baseline ACPs for those cancers had to be adjusted to each individual user based on their current BMI or weight prior to the application of any intervention effects [15,16]. The gene-cancer studies used to estimate the baseline ACPs for CRC and endometrial cancer did not report BMI summary statistics for their patients; therefore, we estimated the mean BMIs for the patients in these studies by sex using other literature sources [4,29,30,33,34]. From there, we were able to adjust the baseline ACPs to an individual based on the difference between their self-reported BMI and the estimated mean BMI for that individual’s sex using Equations (Equation 6)–(Equation 9) for CRC and Equations (Equation 3)–(Equation 5) for endometrial cancer.

*Other intervention assumptions.* Some studies reported risk modifiers that were only linked to a subset of the five LS genes. For these studies, careful consideration went into determining whether or not the results could be generalized to other genes; this was necessary given the lower prevalence of *PMS2* PVs and the limited available literature on LS-related *EPCAM* PVs, as compared to *MLH1*, *MSH2*, and *MSH6* PVs [1,3,4]. For studies where the intervention effects increased over time since the implementation of that intervention, such as for aspirin use, which reported monotonically increasing HRs at 2, 5, and 10 years since implementation, linear interpolation was used to estimate the intervention effects at the years since implementation for which no RR, OR, or HR was reported. The intervention RR, OR, or HR reported in the last follow-up period for an intervention study was assumed to be constant thereafter. For example, the reported HR at 10 years since the start of an aspirin regimen, the last follow-up period reported in the study, was assumed to be constant 11 years post-implementation and afterward [14]. For studies that reported no change in RR, OR, or HR over time, we assumed those values were constant throughout an individual’s lifetime. We also assumed that when multiple interventions were utilized at the same time that their effects were additive due to a lack of studies that explored combinations of interventions.

### 2.3. Programming the Back-End

The interactive, custom, and data-centric concept for the project required a technology stack with (1) a back-end to perform real-time computations on the penetrance data and their modifiers, (2) a UI to allow users to interactively customize and comprehend their risks, and (3) consideration for patient privacy. Therefore, the MyLynch web app was written in R (version 4.1.2) using the Shiny package (version 1.7.3) and deployed to an R Connect server owned by the Dana–Farber Cancer Institute. Shiny web apps become websites once they are deployed, and users can access them from anywhere, so long as they have an internet connection, using a standard web browser via a public website URL. The website uses an https protocol, which protects the user’s data by encrypting the connection between the user’s browser and the server. Shiny web apps, by default, do not store or share any user data with the developer or any third-party. Google Analytics was, and is, used to monitor traffic to the site in order to ensure adequate compute resources and was, and is, configured to collect no other data except site usage over time, by country; this is the bare minimum data that Google Analytics collects. Version control using GitHub made the iterative design process collaborative, trackable, and manageable.

### 2.4. Iterative UI Design

We utilized a two-stage design where the first stage consisted of IRB-approved focus groups led by trained qualitative researchers to solicit feedback before developing a prototype of the tool. After the completion of the focus groups, we developed a prototype of the tool and began the second stage, which consisted of a patient-centric design thinking and usability testing approach, where we iteratively improved the tool based on patient feedback collected during IRB-approved one-on-one cognitive qualitative interviews, also led by trained qualitative researchers [35]. Interview guides were developed by the interdisciplinary team separately for the focus groups and cognitive interviews to ensure these patient interactions were semi-structured [36]. These sessions were conducted over Zoom and recorded for future analysis. The recordings from the focus groups and cognitive interviews were summarized in a de-identified format using a rapid qualitative content analysis approach to understand participant experiences [37]. This process was supported by NVivo software from QSR International.

The inclusion criteria for patients in the focus groups and interviews were the same: (1) a PV in an LS susceptibility gene, (2) at least 18 years of age, and (3) fluent in English. The recruitment pool of patients was identified by screening the Dana–Farber Cancer Institute Cancer Genetics and Prevention clinic schedules and the Progeny and EPIC clinical databases. Patients with and without a previous cancer diagnosis were eligible. Patients participating in the focus groups were excluded from participating in the cognitive interviews. Patients were offered a small monetary incentive, USD 20 in cash, to thank them for participating.

*Focus Groups*. There were four separate focus groups, stratified by age (over/under 40) and cancer history (no cancer history/at least one previous cancer), to ensure varied perspectives, which consisted of two to five patients each. Participants responded to open-ended questions about what they would like in a cancer risk tool for LS patients [36]. Patient perspectives were obtained in the following areas: priorities of content to be included in the tool, risk prevention and mitigation related to LS, formats for receiving risk information, and sharing LS information with at-risk family members.

*Cognitive Interviews.* Based on the focus group results, a prototype was created, which was then subjected to an iterative patient-centric design thinking approach using cognitive interviews in which think-aloud, observational, and retrospective probing usability methods were applied to optimize the web app for its target audience [35,38,39,40]. The cognitive interviews were conducted to obtain patient feedback on ease of use, clarity of information, privacy, user inputs, risk communication visualizations, lifestyle and medical interventions, the social media and email share features, the final report, and the outside resources provided in the web app.

The prototype was presented to 12 recruited, consenting patients for usability testing [41] in individual cognitive interviews between November 2021 and October 2022. During the cognitive interviews, participants were asked to navigate through the content and talk out loud about what they saw, perceived, and thought as they interacted with MyLynch [38,39,40]. A trained member of the research team observed and documented the participants’ responses, interactions with the prototype, and body language [39]. This was followed by an open-ended debrief between the participants and the researcher, which leveraged retrospective probing [40]. The feedback from the interviews was reviewed periodically by the cross-functional research team and was assessed for commonalities, relevance, and feasibility. After each review, the feedback resulted in a vetted list of improvements and new features for the programming team to implement in a test version of the web app, which was then vetted by the research team prior to presentation to the next interviewees.

As an additional harm prevention measure during the cognitive interviews, participants were not permitted to obtain their own personalized risk estimates, given that the web app was still a prototype, and instead were asked to customize risk estimates based on a hypothetical LS patient. A genetic counselor and psychologist were on-call in the event a participant was upset about the LS information or risk estimates they were receiving for the hypothetical patient.

## 3. Results

### 3.1. Baseline ACP Database

Table 2 contains the sources from the literature review that provided gene-cancer risks for different combinations of one of the five LS genes and one of nine cancers: brain, CRC, endometrial, gastric, ovarian, pancreatic, prostate, small intestine, and urinary bladder that occur significantly more frequently in those with LS than those without LS [1,4,28,29,30,33,42,43]. The cancer risks from these studies were stratified by age, gene, and sex from which baseline ACPs were determined for a range of discrete ages from 1 to 85. Where possible, race and ethnicity-stratified risks were also incorporated. These values were stored in our database along with the ACPs from SEER for the general population. The database was configured as an R array by cancer type, genotype, race, ethnicity, and age [26].

### 3.2. Intervention Effects and the Back-End

Table 3 contains a summary of the literature review with the selected intervention studies that cover aspirin use, changes in BMI, colonoscopic screenings, and, for female users, prophylactic surgeries to remove the uterus, both ovaries, and the fallopian tubes [13,14,15,16,17]. The R coded back-end handles the real-time computations that incorporate these risk modifiers by utilizing the database of baseline ACPs, the various RRs, ORs, and HRs identified during the intervention literature review, and the statistical methodology described in the Materials and Methods section.

*Colonoscopy Screenings.* Jarvinen (2020) estimated the RR of CRC for individuals with a *MLH1* or *MSH2* PV who receive colonoscopies every three years, compared to those who do not, as 0.44 [13]. We make the strong assumption that the impact of colonoscopies on CRC risk is the same for all five LS genes. We also assume that the range of colonoscopy frequencies recommended by the U.S.-based National Comprehensive Cancer Network (NCCN) for LS patients (between one to three years, depending on the LS gene), which is similar to other countries’ recommendations, have negligible differences in RR than the value published in the Jarvinen study [44,45]. The app clearly informs the user that their estimate is based on a three-year colonoscopy frequency and that their clinician may recommend a different frequency. The reported RR value could not be applied directly to the baseline ACPs because the studies in Table 2 did not control for colonoscopies among their samples of LS patients [4,29,30]. These patients were a mix of US, Canadian, Western European, and Australasians [4,29,30]. We estimated the proportion of the sample of people represented in our baseline ACPs who underwent regular colonoscopies at some prescribed interval to be 0.818, which was based on two studies on colonoscopies among LS patients, or patients exhibiting characteristics similar to LS patients, in the US and Australasia, respectively [31,32]. Using this proportion, we were able to obtain adjusted ACPs for CRC representing individuals who have not undergone regular colonoscopies using Equation (Equation 5). When these adjusted baseline ACPs corresponded to absolute lifetime risks above 90%, then these penetrance estimates were capped so that the lifetime absolute risks reached a maximum of 90%. This adjustment brought these risks in-line with other published literature [46,47]. The RR from the Jarvinen study could then be applied to those adjusted baseline ACPs to obtain post-intervention ACPs for patients who underwent regular colonoscopies.

*Aspirin.* A randomized controlled trial by Burn (2020) estimated a CRC HR for individuals with an *MLH1*, *MSH2*, or *MSH6* PV taking 600mg of aspirin per day, compared with those not taking aspirin, to be 0.56 at 2 years from the start of the intervention, 0.63 at 5 years, and 0.65 at 10 years [14]. These three estimates were used to smooth and extrapolate a series of HRs for modifying the baseline ACPs. We made the assumption that these estimates can also be applied to individuals with *PMS2* and *EPCAM* PVs. We also assumed that the samples of patients from the CRC studies in Table 2, which were published between 2011 and 2015 and which we used to establish baseline ACPs, consisted of an insignificant number of individuals on a high-dose aspirin regimen due to the recency of the 2015 aspirin finding. The app clearly informs the user that their CRC risk estimate while using aspirin is based on a 600mg dosage but that their clinician may recommend a different dosage or no aspirin at all.

*BMI.* Research by Movahedi (2015) indicates each single point increase in BMI, measured in kg/m${}^{2}$, increases the HR of CRC by 7% over five years for individuals with a *MLH1* PV [15]. Similarly, work by Trentham-Dietz (2006) indicates for each 5 kg weight gain, the OR of women developing endometrial cancer increases by 20% in the general population, and we assumed that this OR applies to LS patients [16]. We assumed that weight loss and decreasing one’s BMI lowers CRC risk for those with a *MLH1* PV and lowers endometrial cancer risks for females with a PV in any LS gene with an endometrial cancer association, and we assumed this risk reduction corresponds to the same HR and OR listed in the Movahedi and Trentham-Dietz studies for increases in BMI and weight, respectively. The app requests each user’s height and weight so we are able to convert between weight and BMI as needed. The baseline ACPs for these two cancers were based on studies that sampled patients from the US, Canada, Western Europe, and Australasia; therefore, we assumed that our penetrance estimates, which were conditional on sex, represented a mean of all the national average BMIs from the countries included in the studies, by sex, which was 27.1 kg/m${}^{2}$ for males and 26.0 kg/m${}^{2}$ for females [34]. Based on the difference between a user’s BMI and the multi-country mean BMI for their sex, the baseline ACPs are adjusted using the HR for CRC or the OR for endometrial cancer that corresponds to that BMI difference using Equation (Equation 9). For users in or below the normal BMI range (<25 kg/m${}^{2}$), their baseline ACPs are only adjusted to correspond to a BMI of 25 kg/m${}^{2}$ and, for obese users (a BMI of ≥30 kg/m${}^{2}$), their baseline ACPs are adjusted to correspond to a maximum BMI of 30 kg/m${}^{2}$; these caps prevent a user’s current BMI from having an unrealistic impact on their initial cancer risks. In the web app, if the user’s current height and weight correspond to a BMI above 25 kg/m${}^{2}$, a range of weight loss options is provided to the user, in increments of one pound, from zero pounds to the number of pounds equivalent to either five BMI points or the minimum BMI reduction needed to reach the normal range, whichever is smaller. This range was established to prevent encouraging a user from losing weight if they already have a healthy BMI, and it ensures that the effect of weight loss on cancer risks is not overstated for very obese people who desire to lose a lot of weight.

*Prophylactic surgeries.* Prophylactic hysterectomies (removal of the uterus) and prophylactic salpingo-oophorectomies (removal of both ovaries and the fallopian tubes) are somewhat common means for preventing endometrial and ovarian cancer from developing in patients with a strong family history of cancer [17]. We assumed that both surgeries reduce the cancer risk for the relevant organs down to zero in LS patients [17].

### 3.3. Results from the Focus Groups and Resulting Initial Prototype

The focus groups were conducted between October 2020 and March 2021. Seventeen patients were recruited and consented to the focus groups; however, only 14 participated due to scheduling constraints. There were two participants in the ‘less than age 40 with no previous cancers’ group, four in the ‘age 40 or older with no previous cancers’ group, three in the ‘less than age 40 with a previous cancer’ group, and five in the ‘40 or older with a previous cancer’ group. Participants with PVs in all five LS genes were represented and approximately half were female and half male. The mean age was 49.6 (approximate range ≥18 and <80) and the majority were White, non-Hispanic, and without Ashkenazi Jewish ancestry. The mean number of primary cancers among those with a personal history of cancer was 1.5, the mean number of relatives with a cancer history was 6.0, and the mean number of relatives with LS was 2.5.

The focus groups provided similar feedback to one another, and a summary of this feedback is found in Table 4. The main observations were that participants wanted a simple, interactive tool with intervention options and that the provided intervention options would make them feel hopeful and empowered. They also desired a solution that would provide information in a variety of ways for different personal learning preferences. Based on this general feedback, we developed an initial prototype, a screenshot of which can be found in Appendix A.

### 3.4. Results from the Cognitive Interviews and Iterative UI Design

*Cognitive Interviews Summary.* Among the 12 consented patients interviewed, individuals with PVs in all five LS genes were represented, approximately half were female and half male, the mean age was 53.8 (approximate range ≥18 and <80) and the majority were White, non-Hispanic, and without Ashkenazi Jewish ancestry. The mean number of primary cancers among those with a personal history of cancers was 1.8, the mean number of relatives with a cancer history was 2.0, and the mean number of relatives with LS was 2.2.

A summary of the feedback from the 12 cognitive interviews is found in Table 5. The participants’ overall impression of the web app was very positive, even in early versions, with participants reporting that it was informative and concise, empowering, easy to use and navigate, and anxiety-reducing. Almost all said they would share MyLynch with their family, and some reported they preferred the web app over receiving the information from their clinician or genetic counselor.

The most common concerns, which were addressed during the iterative design process, were the clarity and quality of instructions and content, navigation difficulties, visual appearance, explanation of the interventions, and privacy. Although not currently feasible due to a lack of studies specific to LS patients and compatible with our database, several patients suggested including more intervention options such as diet and exercise. Additionally, many interviewees pointed out that people may not know their LS gene, which would make using the web app difficult.

*Iterative UI design based on the cognitive interviews.* As a result of the cognitive interviews, the web app went through significant, positive changes in content and appearance. We found that a homepage with general information about LS stating the purpose of the web app improved user satisfaction, likely because it added valuable context (see Appendix A). Importantly, to address participant concerns, the homepage prominently and concisely summarizes the privacy policy by stating that the personal information the user enters will not be shared or stored. The language used in the web app was carefully selected to cater to audiences without a medical or quantitative background, and after multiple iterations of the web app, we found it was most effective to break the information into small, progressive steps with repetition of information to reinforce learning. The progress bar displayed at the top of each screen allows the user to keep track of these steps as they navigate through them. A flow chart of the user’s journey through MyLynch is found in Figure 1. During the step in which the user enters their personal information, depicted in Appendix A, special care was taken to explain the need for each user input by using a combination of annotations and hover text (see Appendix A). Because the visualizations may be overwhelming initially, we preceded them with a page containing a table reporting lifetime absolute and relative cancer risks, as shown in Appendix A, to ease the user into the concepts being conveyed.

It is generally understood, and supported by opinions expressed in our focus groups and cognitive interviews, that learning styles vary person-to-person based on preference and graphical and numerical literacy; therefore, users are presented with examples of three different visualization styles, a line graph, a bar graph, and a personograph, and asked for their preference [9,10,12]. Figure 2 provides examples of each visualization style, all conveying the same risk information. The line graph provides the most detailed information by showing the user cumulative risks, rounded to the nearest 1%, for each year between their current age and age 85. The bar graph provides just two focused snapshots of risk within five years and by age 85, also rounded to the nearest 1% [10]. The personograph, an info-graphic for communicating part-to-whole ratios using people-shaped icons with different colors or shading, provides the same two snapshots of risk during the user’s lifetime as the bar graph; however, the final versions of our personographs had only 20 people icons, so the granularity of the risk is limited to increments of 5%. We found that more than 20 icons were visually overwhelming when comparing several adjacent personographs while fewer icons provided too little granularity to appreciate the magnitude of the impact of the interventions. In all three options, each cancer has its own unique color, and the average person without LS is represented by the neutral color grey for contrast. For users who may have difficulty discerning colors, we utilized labels and different line and bar patterns to differentiate cancer risk estimates.

When the user toggles these different interventions on and off using the visualizations’ interactive features on the side menu, the plot in the center of the screen changes simultaneously to demonstrate how their cumulative risks could change if the user were to adopt the selected interventions. The line, bar, or personograph that represents their cumulative risks with no interventions remains fixed on the screen at all times, as does the line, bar, or personograph representing the cumulative cancer risk for the average person, while a third line, bar, or personograph moves interactively as the user selects the different interventions. We found that the patients liked seeing how the interventions lowered their risks, but early versions of the web app lacked a thorough explanation of the interventions. Therefore, the screen depicted in Appendix A was created with this information.

After viewing and interacting with the visualizations, the user is given the opportunity to customize a PDF report and then download or email that report. The page instructions recommend that the user sends it to themselves, their primary care provider, and/or their genetics provider. This was a popular feature in the interviews and is key for enhancing patient–provider communication. Figure 3 shows the interactive UI with which the user builds their personalized cancer risk report.

Although the majority of the interviewees reported they would share the website, it was also observed that many seemed unaware of the share buttons in the upper right-hand corner; therefore, a final step was added to the web app, which attempts to highlight the potential benefits to the user’s own family if the user were to share MyLynch and/or their LS diagnosis; a screenshot of this page is shown in Appendix A. The benefits are explained by presenting facts on (1) the importance and effectiveness of early diagnosis and screening, (2) the large portion of people with LS who are undiagnosed, and (3) the extent to which their first-degree and more distant relatives are at risk. The screen also provides resources for how to go about sharing an LS diagnosis with family members, and finally, it prominently displays large versions of the social media and email share buttons for sharing the link to MyLynch.

## 4. Discussion

The main result of this research was the MyLynch web app for LS patients (https://MyLynch.org) (accessed on 2 December 2022); it received very positive patient feedback overall. MyLynch leads the user through a series of steps with dynamic inputs that personalize their cancer risks. The user’s inputs also generate a unique menu of lifestyle and medical interventions available to them. The risks are provided in three different user-friendly visualization styles, based on preference, and allow the user to simultaneously comprehend and compare their risks for each cancer to the general population, their risks over time, cancers relative to one another, and the impact of interventions. The web app also provides a PDF report that the user can send to their clinician. Finally, the user is provided with information to encourage disclosure of their diagnosis with family members, which can be easily achieved using in-app buttons.

Based on using MyLynch and consulting their clinician, the user is well informed and enabled to make lifestyle and medical intervention decisions best suited to their needs and in consultation with their clinician. Early diagnosis of LS allows medical providers to prescribe tailored screening and interventions that are highly effective in preventing cancer and, at least for colonoscopies, extending life expectancy [48,49]. MyLynch aims to help patients understand that LS-related cancers are preventable and manageable by highlighting the impact of each intervention.

The interactive visualizations in MyLynch are the primary modes of risk conveyance to the user. Each of the visualization options utilized by MyLynch has its strengths and weaknesses, detailed below, but all three clearly communicate four things to the user: (1) their cancer risk and how it changes throughout their lifetime, (2) how their risk compares to someone without LS, (3) how their risk differs between types of cancer, and (4) how potential interventions would change those risks. The changes in risk levels over a user’s lifetime can inform patients and their clinicians on the optimal timing for certain interventions, while the between-cancer comparisons allow for prioritization of interventions by cancer type. The comparisons of risks to someone without LS provides context and possibly motivation to seek out medical advice and interventions. Line graphs are best at showing trends in a person’s risk over time. For example, a user will be able to see at what age their risk levels begin to either accelerate or taper off, which provides context to assist with decisions about when to implement invasive interventions such as prophylactic surgery [10]. Although the ability to see trends is much more limited using the bar graph, evidence suggests they are very effective at communicating differences in probabilities between scenarios, such as differences between a user’s risk with and without interventions, because the brain will naturally subtract the heights of the bars to comprehend the difference [10]. Although the personograph provides the least granularity of cancer risks, the literature suggests it is effective for risk comprehension and recall of information for users with both high and low numeric literacy [10,11,12,50]. The line and bar graphs are more compact; therefore, different cancer types can be compared simultaneously in separate facets of the same visualization, whereas in the personograph, the user must toggle through the different cancers one at a time. Regardless of the patients’ preferred visualization style in the cognitive interviews, they were found to be effective conveyors of risk information. There was no one favorite visualization style amongst the patients, with each style being preferred by at least some of the interviewees, confirming the need to provide several options tailored to each user.

Cascade testing for at-risk relatives is usually recommended for LS families due to the heritable nature of the syndrome [48,51] and, LS diagnosis disclosure within families is critical for identifying untested family members because many people would otherwise be unaware of the opportunity to receive germline genetic testing [52]. Although most LS patients share their diagnosis with some family members, patients may be hesitant to do so with all at-risk family members, in part because they do not want their family members to worry and/or due to a perception that the family member may not understand the condition [53]. According to our focus groups and interviews, testing resistance by informed relatives is partially due to a poor understanding of the risks associated with LS and a misconception that there is little to be done to mitigate these risks. MyLynch may increase cascade testing by mitigating the concerns of both patients and their family members.

The rapidly growing body of knowledge of genetics, a limited supply of knowledgeable medical professionals, such as genetic counselors, resource constraints of general practitioners, and the recent and rapid increase in access to genetic testing at reduced costs may make it increasingly difficult for LS patients to receive counseling on the most up-to-date risk estimates for the variety of different cancers for which they are at risk [6,18,54]. MyLynch has the ability to become a force multiplier, which can aid providers in counseling patients and providing hope.

MyLynch has several limitations, which stem from a lack of available data. For example, certain cancers such as hepatobiliary, sebaceous carcinoma, kidney, and ureter have been associated with LS; however, we lack enough data on these cancers’ associations with LS genes to estimate ACPs by at least age, sex, and gene [55,56]. Additionally, some cancers may have a significantly increased risk associated with multiple LS genes; however, we may only have enough data to estimate ACPs by age, sex, and gene for a subset of those associated genes [1,55]. For example, gastric cancer may have significantly increased risk for carriers of *PMS2* PVs but based on the lower prevalence of *PMS2* PVs and less published literature on *PMS2*, as compared to the other LS genes, we can only estimate ACPs for gastric cancer that are associated with *MLH1*, *MSH2*, and *MSH6* by age, sex, and gene [1,3]. There is also limited literature available on cancer risks for individuals with PVs in two different LS genes; therefore, MyLynch does not provide tailored risk estimates specifically for these individuals. However, we recommend that these patients compare the future risk estimates, by cancer type, for each of their PV carrying genes and that they assume the highest risk-conferring gene among the two provides the best estimation of their personalized cancer risk. MyLynch does not currently assess the risk of recurrence of the same cancer type, again due to a lack of specific studies that provide these risks for patients with LS in enough detail for us to generate ACPs.

Regarding limitations for estimating the intervention effects, there were strong assumptions made due to a lack of specific, detailed studies with statistically significant results. For example, interventions may combine in ways that are unlikely to be perfectly additive; however, due to a lack of data on these interactions, we chose to assume they have an additive effect when combined. The Movahedi (2015) study did, however, report an increased and significant HR for obese subjects in the aspirin placebo group while also reporting an increased but insignificant HR for obese people in the aspirin group [15]. Furthermore, aspirin effects were estimated in a study using a 600mg daily dose [14]; however, some clinicians will not feel comfortable prescribing this dosage due to toxicity concerns. Similarly, the study used to establish the RR for colonoscopies was based on a three-year interval between screenings, but these intervals can vary based on the guidelines of a patient’s country of residence, their age, and other factors [13,44,45]. For example, the NCCN recommends LS patients receive a colonoscopy every 1 to 3 years starting between age 20 to 35 or 2 to 5 years before the youngest case of CRC in the family depending on which gene has a PV [44]. Due to the fact that the studies we used to estimate the baseline ACPs did not control for many of the interventions, we also adjusted, or chose not to adjust, these penetrance estimates based on assumptions made about the sample of subjects used in those studies; including assumptions on their aspirin use, average BMI, and the proportion of subjects who follow a regular colonoscopy schedule [13,14,15,16,17].

Risk uncertainty was not communicated to the users of MyLynch because this information could have created additional risk communication complexities, which were beyond the scope of this work. These complexities include how personality traits, cognitive attributes, and motivational factors of information recipients affect their perception of risk models and their personal risk estimates [57]. Due to these factors, inclusion of risk uncertainty for some information recipients has been associated with feelings of worry, distrust of, or disinterest in risk models, and individuals provided with the same risk uncertainty information have been shown to have large variations in perceived personal risk [57].

We have several future improvements planned for MyLynch, including the development of a mobile device compatible version optimized for viewing on a smaller but commonly used screen size. This could make MyLynch more accessible to under-served populations who are more likely to utilize a smart phone to access health information and resources [58]. In the future, we are planning to augment the web app with an artificial intelligence relational agent to improve patient interaction. Additional validation studies using larger patient samples are needed to assess the accuracy of our risk estimates and to assess MyLynch’s effects on cascade testing rates and its ability to influence clinical decisions. We plan to update the tool as new literature is published on LS gene-cancer associations and interventions for LS patients. These updates may include uncertainty information. A user survey using REDCap, a secure survey and database tool, was added to the end of MyLynch to capture additional user feedback for continuous improvement. The methods used to create this web app can also be extended to benefit patients of other, non-LS, inherited cancer syndromes such as hereditary breast and ovarian cancer (HBOC), Peutz–Jeghers syndrome, and more.

## 5. Conclusions

As germline genetic panel testing becomes more widely available, GPM will play an increasingly important role in patient care, and CDS tools offer patients and providers tailored information to inform decision-making [6,22,23]. MyLynch, a GPM CDS tool and web app, provides LS patients with personalized, gene-specific cancer risks; educates patients on relevant interventions; and provides patients with adjusted risk estimates, depending on the interventions they choose to pursue. The back-end of the app was built around a database of ACPs of gene-cancer associations, covering all five LS genes and nine different cancer types. These ACPs and the intervention effects for colonoscopies, aspirin, weight loss, oophorectomies, and hysterectomies were estimated using a rigorously curated literature review and the application of the developed statistical methods. The UI was iteratively developed based on patient feedback from 4 focus groups and 12 cognitive interviews and yielded significant improvements to the usability of the web app. MyLynch received very positive patient feedback and has the potential to improve clinician–patient communication and encourage adoption of risk-reducing strategies and cascade testing. The MyLynch risk estimates and their clinical utility should be validated on larger samples of patients in future studies. We also plan to produce a mobile compatible version of the app as well as incorporate risk uncertainty into the estimates.

## Figures and Tables

**Figure 1 cancers-15-00391-f001:**
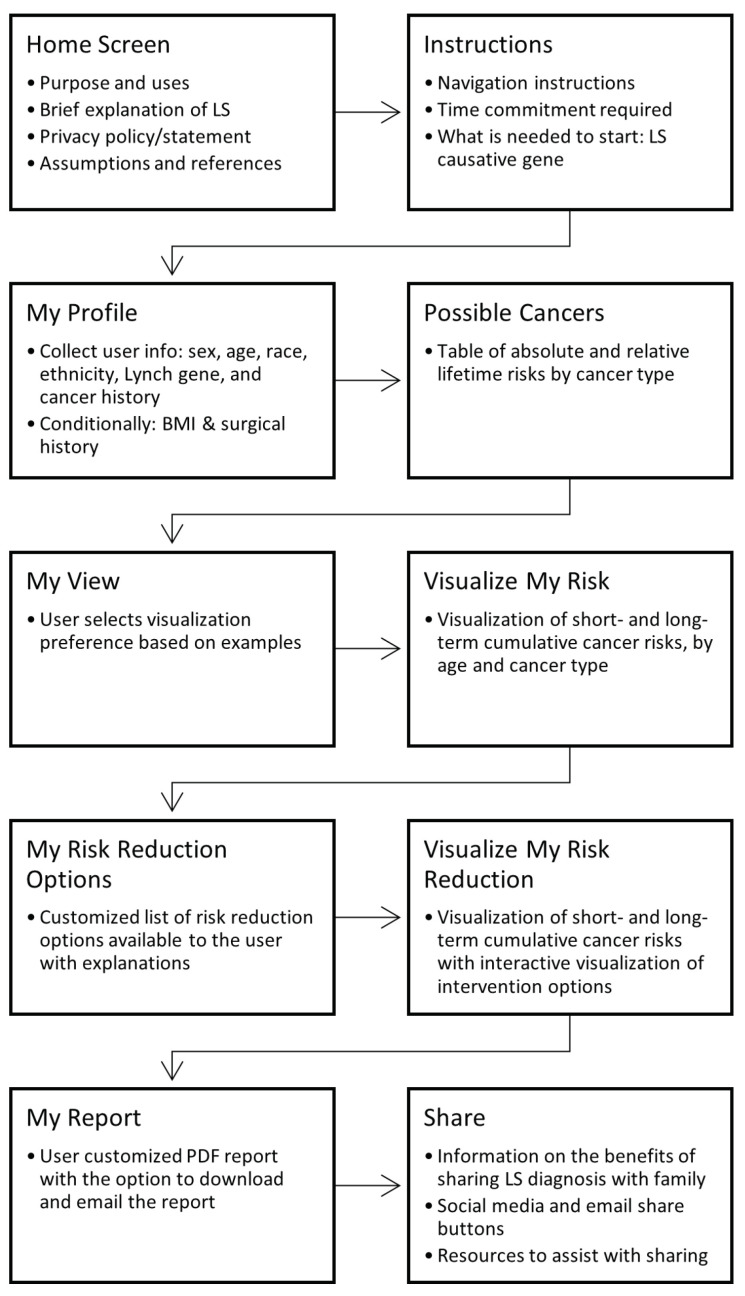
Flowchart of steps a user takes to navigate MyLynch.

**Figure 2 cancers-15-00391-f002:**
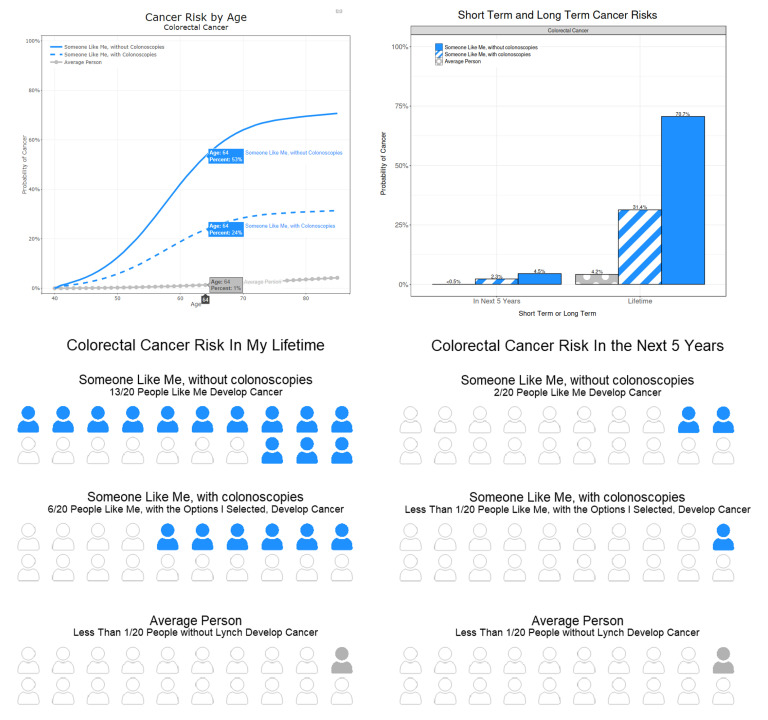
Three visualization options available to the user, a line graph (top left), a bar graph (top right), and personographs (bottom), each conveying the same information: the CRC risk over time, with and without regular colonoscopies, for a hypothetical 40-year old male, with an unspecified race and ethnicity, a *MSH6* PV and no previous cancer history.

**Figure 3 cancers-15-00391-f003:**
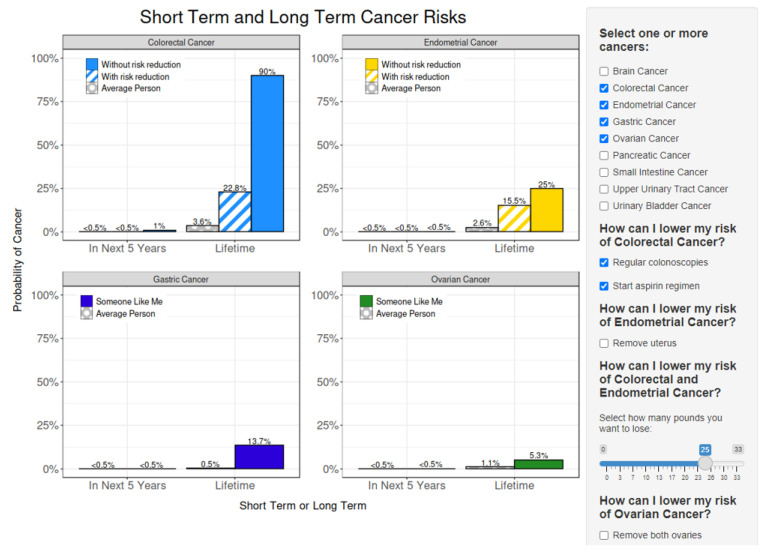
Screen shot from MyLynch in which the user builds their personalized cancer risk report using the bar graph visualization. The user selects options on the right, which interactively adjusts the risk visualization on the left. The figure shows CRC, endometrial, gastric, and ovarian cancer risks over time for a hypothetical 25-year-old, obese, female with an unspecified race and ethnicity, a *MLH1* PV, no cancer history, and no relevant prophylactic surgical history. The hypothetical user has selected the following interventions: regular colonoscopies, an aspirin regimen, and 25lbs of weight loss.

**Table 1 cancers-15-00391-t001:** Notation used to describe our approach to the calculation of risk, considering one cancer type at a time.

Notation	Interpretation
*c*	A cancer type where *c* is one of the cancer types listed in Table 2.
$T_{c}$	The age of occurrence for cancer *c*.
$G_{j},g$	$G_{j}$ is the genotype of gene *j*, where j=(*MLH1, MSH2, MSH6, PMS2, EPCAM*). *g* is a binary indicator of the presence of a PV in gene *j*. When $G_{j}=1$ a PV in gene *j* is present and when $G_{j}=0$ a PV in gene *j* is absent.
S,s	*S* is the sex where *s* can be either male or female.
R,r	*R* is the race where *r* is either All_Races, for individuals with an unknown race or who are mixed race; American Indian/Alaskan Native (AIAN); Asian/Pacific Islander (API); Black; or White.
E,e	*E* is the ethnicity where *e* is either All_Ethnicities, for individuals with an unknown ethnicity or a mix of Hispanic and non-Hispanic ethnicity; Hispanic; or non-Hispanic.
$a_{cur}$	The individual’s current age, in discrete units (e.g., years).
$\left(a_{cur},a_{fut}\right]$	The age interval from the individual’s current age $a_{cur}$ to some future age $a_{fut}$ for which the cumulative risk is to be estimated and where $a_{fut}>a_{cur}$.
$S_{c,g,s,r,e}\left(a_{cur}\right)$	The survival probability of living free from cancer *c* until the individual’s current age $a_{cur}$ given genotype *g* for gene *j*, sex *s*, race *r*, and ethnicity *e*.
$\Phi_{k},k$	$\Phi_{k}$ is a binary indicator of whether intervention type *k* was applied or not, where *k* is one of the interventions listed in Table 3. $\Phi_{k}=1$ when the intervention *k* is applied and $\Phi_{k}=0$ when it is not.
$RR_{c,a,g,s,r,e,k}$	The effective RR for intervention *k* for cancer *c*, at age *a*, given genotype *g* for gene *j*, sex *s*, race *r*, and ethnicity *e*.
$OR_{c,a,g,s,r,e,k}$	The effective OR for an intervention *k* for cancer *c*, at age *a*, given genotype *g* for gene *j*, sex *s*, race *r*, and ethnicity *e*. If the intervention has a binary application type (e.g., a patient either receives regular colonoscopies or not), then this is the reported OR from an intervention study.
$|\Delta v_{k}|,\sigma_{v_{k}}$	$|\Delta v_{k}|$ is the absolute value of the change in the continuous variable *v* used to measure the application of intervention *k* (e.g., amount of change in body weight). $\sigma_{v_{k}}$ is a dichotomous variable that indicates whether $|\Delta v_{k}|$ will have an increasing or decreasing effect on cancer risk based on the study finding for intervention *k*. $\sigma_{v_{k}}=1$ if $|\Delta v_{k}|$ increases cancer risk and $\sigma_{v_{k}}=-1$ if $|\Delta v_{k}|$ decreases cancer risk. These variables are for use with interventions that are applied on a continuous scale only.
$OR_{c,a,g,s,r,e,k,v_{k}}$	The reported OR for an intervention *k*, which is applied on a continuous scale measured by $v_{k}$ (e.g., change in body weight), for cancer *c*, at age *a*, given genotype *g* for gene *j*, sex *s*, race *r*, and ethnicity *e*.
$\lambda_{c,a,g,s,r,e,\Phi_{k}}$	The hazard rate for cancer *c*, at age *a*, given genotype *g* for gene *j*, sex *s*, race *r*, ethnicity *e*, with intervention *k* either applied or not applied.
$HR_{c,a,g,s,r,e,k}$	The effective hazard ratio (HR) for an intervention *k* for cancer *c*, at age *a*, given genotype *g* for gene *j*, sex *s*, race *r*, and ethnicity *e*. If the intervention has a binary application type (e.g., a patient either adheres to an aspirin regimen or not) then this is the reported HR from an intervention study.
$HR_{c,a,g,s,r,e,i,v_{k}}$	The reported hazard ratio (HR) for an intervention *k*, which is applied on a continuous scale measured by $v_{k}$ (e.g., change in BMI), for cancer *c*, at age *a*, given genotype *g* for gene *j*, sex *s*, race *r*, and ethnicity *e*.
$p_{c,a,g,s,r,e,k}$	The estimated proportion of patients in a study sample who adhered to intervention *k* for cancer *c*, at age *a*, given genotype *g* for gene *j*, sex *s*, race *r*, and ethnicity *e*. $\left(1-p_{c,a,g,s,r,e,k}\right)$ is the estimated proportion of patients who did not adhere to intervention *k*.

**Table 2 cancers-15-00391-t002:** References used to estimate the baseline ACPs for each gene-cancer association. Blank cells indicate that no association between a gene and cancer was incorporated into MyLynch due to a lack of sources that could demonstrate individuals with a PV in the corresponding gene were at significantly higher risk for the corresponding cancer than someone without a PV in that gene. Refer to the following references for the sources in this table: Dominguez-Valentin (2020) [1], Dowty (2013) [28], Engel (2012) [42], Felton (2007) [33], Kempers (2011) [4], Møller (2018) [43], ten Broeke (2015) [30], and Wang (2020) [29].

Cancer Type		Gene with Pathogenic Variant				
		*MLH1*	*MSH2*	*MSH6*	*PMS2*	*EPCAM*
**1**	**Brain**	Møller (2018)	Møller (2018)	Møller (2018)		
**2**	**CRC**	Wang (2020)	Wang (2020)	Wang (2020)	ten Broeke (2015)	Kempers (2011)
**3**	**Endometrial**	Felton (2007)	Felton (2007)	Felton (2007)		
**4**	**Gastric**	Dowty (2013)	Dowty (2013)	Møller (2018)		
**5**	**Ovarian**	Engel (2012)	Engel (2012)	Møller (2018)	Engel (2012)	
**6**	**Pancreatic**	Møller (2018)	Dowty (2013)			
**7**	**Prostate**		Dominguez-Valentin (2020)			
**8**	**Small intestine**	Engel (2012)	Engel (2012)	Engel (2012)		
**9**	**Urinary bladder**	Møller (2018)	Møller (2018)	Møller (2018)		

**Table 3 cancers-15-00391-t003:** References used to estimate the effects of the interventions that were incorporated into MyLynch, by cancer type, with the significant RR, OR, and HR values and the LS genes from those studies. We assumed some RRs, ORs, or HRs for the interventions were applicable to LS genes not included in each study; see the Materials and Methods section for details. Refer to the following references for the sources in this table: Burn (2020) [14], Jarvinen (2020) [13], Movahedi (2015) [15], Schmeler (2006) [17], Trentham-Dietz (2006) [16].

Cancer Intervention	RR, OR, or HR Utilized	Reference	Genes Supported in the Study
**CRC**			
Colonoscopies	RR: 0.44 for screening every 3 years	Jarvinen (2020)	*MLH1, MSH2*
Aspirin Regimen	HR: 0.56 2 years after initiation HR: 0.63 5 years after initiation HR: 0.65 10 years after initiation all HR were for a 600MG dose/day	Burn (2020)	*MLH1, MSH2, MSH6*
Lower BMI	HR: decrease of 7% for each 1 point decrease of BMI	Movahedi (2015)	*MLH1*
**Endometrial**			
Weight Loss	OR: decrease of 20% for each 5kg in weight lost in overweight and obese patients	Trentham-Dietz (2006)	This study was of the general population and was not specific to LS patients.
Prophylactic Hysterectomy	No occurrences of women with endometrial cancer after surgery; equivalent to RR = 0	Schmeler (2006)	*MLH1, MSH2, MSH6*
**Ovarian**			
Prophylactic Oophorectomy	No occurrences of women with ovarian cancer after surgery; equivalent to RR = 0	Schmeler (2006)	*MLH1, MSH2, MSH6*

**Table 4 cancers-15-00391-t004:** Summary of identified domains in the focus groups.

Domain	Focus Group Summary
(1) Overall Suggestions	Provide a range information and resources in one place Easy to share with family and care team Simple and easy to navigate Optimistic, hopeful, and empowering Address privacy and information security concerns
(2) Priority Information	Personalize the information Make the information actionable Focus on risk reduction Emphasize cancer risk over lifetime
(3) Visualization Preferences	Include a variety of visual options to account for different personal preferences Include comparisons to people without LS Interactive charts and graphs Colorful

**Table 5 cancers-15-00391-t005:** Summary of identified domains in the cognitive interviews.

Interview Summaries by Domain	Representative Quotes	Related Features
**Overall Impression**		
**Positive Feedback:** Experience was calming and relieving because you can see what you are doing to reduce your cancer risk A LS diagnosis is overwhelming but, the tool was useful and easier than getting results from your doctor Put things into a realistic perspective Simple, straightforward, interactive, personalized, and had good instructions Helps people understand risk lowering activities and reassures them, and even motivates them to do it, because they can see it makes a difference Allows people to be more informed and take that information to their doctor	“I was a little nervous of seeing this [website] before we started... but when you show it like this, it just makes me feel more relaxed and at ease” “I wish I had something like that when I was diagnosed”	
**Ease of Use and Clarify of Information**		
**Positive Feedback:** Information was clear, concise, helpful, and easy to read and understand The step-by-step nature of the tool made it easy to navigate, good length, and minimal effort required **Addressed Concerns and Suggestions:** Description of tool is unclear; reference to ‘risk’ is unclear Make the different tabs clearer and emphasize how to navigate through the site with a directional arrow Make each page read left to right; reading portions on left and action portions on the right **In Progress:** Make it mobile compatible	“Looks so much better than the information that I got from the genetics center where I was diagnosed…” “It’s easy on the eyes, it’s easy on the language…. and it’s familiar”	Carefully selected language, written for a diverse audience Step-by-step navigation Home page Instruction screen
**User Inputs**		
**Positive Feedback:** It was easy to put in the information and was similar to what you would give at the doctors The black pop-up information boxes were very helpful Appreciation of the breakdown by mutation type **Addressed Concerns and Suggestions:** The scale/ruler for age and weight are a bit cumbersome People might not know their problem gene Don’t default weight to 100lbs because it is emotional to move up from there Can gender options be made more inclusive? Some might be confused and think the example visualizations are their own data Why isn’t race/ethnicity included?		Dedicated screen for user inputs Minimal collection of info Details provided with hover text and annotations Selection of preferred visualization style with examples
**Privacy**		
**Positive Feedback:** Inclusion of disclaimer and terms/conditions **Addressed Concerns and Suggestions:** Make it clear that your information will not be saved or seen by others Mixed perspective on privacy: some no concern others want clear statement on privacy		Privacy statement No data sharing or storing https encryption Terms and conditions
**Risk Conveyance Table, Visualizations, and Aesthetics**		
**Positive Feedback:** Good, clear, helpful visualization of risk and risk mitigation. It put risk in perspective Includes a variety of visual options to account for different personal preferences Inclusion of many different cancers Comparison to general population Positive perception of the risk table of cancers **Addressed Concerns and Suggestions:** Need more vivid, contrasting colors on the graphs Mixed views and confusion on long-, mid-, and short-term comparison and their labels How is the “average” person defined? What data was used to make the graphs? In the personograph, why is the average person always black? The homepage only has plain text and no visuals; add more infographics, colors, and text effects Changing y-axis maximum height when switching between views is confusing **Unaddressed Suggestions for Improvement:** Include all possible cancers and cancers the user has already had	“I like that people who visualize data differently can choose” “I love that you put [cancer risk] into [terms of] 22 times more risk [than the average person] because a lot of the medical places will just say your lifetime risk and the average person’s lifetimes risk and then you have to kinda compare that... a lot of people who don’t have a background in statistics would understand [22 times more risk]”	Visualization style options Visualization color scheme Comparisons to someone without LS, different cancers, several time points Table with lifetime absolute and relative risks PDF document with supporting assumptions and references
**Interventions**		
**Positive Feedback:** Helps people understand available risk lowering activities, reassures them, and motivates them to do it because they can see it makes a difference Helpful inclusion of the comparison of CRC risk with and without colonoscopies Helps to prioritize; you understand why you get some screenings but don’t need a brain MRI every year **Addressed Concerns and Suggestions:** Explanations of intervention scenarios unclear Include more risk reduction strategies like diet, weight loss, smoking, and exercise Confusion on the recommended frequency of colonoscopies Explain why some people might have different risk reducing measures **Unaddressed Suggestions for Improvement:** When there isn’t a drastic change, I am less inclined to feel like the risk reduction is worth it	“If I take these things seriously, I can really reduce my risk” “…a colonoscopy is not pleasant, but if I can see that it would really make a difference, I would do it” “Something that you can use to understand the risk and understand how preventable some of these things can be”	Interactive visualization of intervention effects Practical risk mitigating interventions Dedicated screen for interventions explanations
**Sharing and the Report**		
**Positive Feedback:** Helpful to be able to download the report and bring it to your doctors. This is important. Would recommend the website to family Would share the report and the tool with family to encourage them to get tested **Addressed Concerns and Suggestions:** Make share buttons more visible and colorful	“My other three siblings… still have to be tested and two of them have children, and I think it is a helpful way to use it as an education material for people”	PDF report Share buttons Dedicated screen to encourage disclosure of diagnosis
**Outside Resources**		
**Positive Feedback:** Love inclusion of other places to get more information **Addressed Concerns and Suggestions:** Include links to additional information and resources on who to contact such as a genetic counselor		Dana-Farber LS link ACS link KinTalk.org link

## Data Availability

The focus group and interview video recordings of the patients are personally identifiable information and cannot be made public due to privacy.

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
