# Peer review of "MyLynch: A Patient-Facing Clinical Decision Support Tool for Genetically-Guided Personalized Medicine in Lynch Syndrome"

_cancers, 2023, doi:10.3390/cancers15020391_

Round 1

Reviewer 1 Report

Paper deals with important task in Personalized Medicine. The authors developed a patient-facing clinical decision support  web application that applies genetically-guided personalized medicine for individuals with Lynch syndrome.

Paper has great practical value.

It has a logical structure all necessary sections. The paper is technically sound. The experimental section is very good.

Suggestions:

1.       It would be good to add point-by-point the main contributions at the end of the Introduction section

2.       It would be good to add the remainder of this paper at the end of the Introduction section

3.       It is not clear from the paper what scientific results are included in the development web application. It would also be good to show them in the main contribution of the article.Authors should provide a link to open access repository with the dataset used for modeling

4.       The conclusion section should be extended using: 1) numerical results obtained in the paper; 2) limitations of the proposed solution; 3) prospects for future research.

5.       A lot of references are outdated. Please fix it using 3-5 years old papers in high-impact journals.

Author Response

Point 1: It would be good to add point-by-point the main contributions at the end of the Introduction section

Response 1: We thank the reviewer for their thoughtful comment. The point-by-point contribution of the app are summarized in this new paragraph at the end of the Introduction section:

MyLynch provides LS patients with personalized cancer risks based on their unique profiles and educates patients on available interventions and the extent to which those interventions can lower their cancer risks. MyLynch was well received by the patients in our focus groups and interviews and has the potential to improve communication between patients and clinicians, as well as encourage patients and their family members to adopt risk-reducing strategies and undergo cascade testing.

Point 2: It would be good to add the remainder of this paper at the end of the Introduction section

Response 2: We agree that adding text to detail the remainder of the paper would benefit readers by helping them to understand the outline of the paper. The following paragraph, which summarizes the remainder of the paper,  was added to the end of the Introduction section:

In the Materials & Methods section, we will describe the literature review and statistical techniques used to estimate personalized cancer risks, with and without risk-reducing interventions, as well as the design of the patient focus groups and cognitive interviews used to develop the web app. In the Results section, we will summarize the database of cancer penetrances, which incorporated the five LS genes and nine different cancer types, that was used to estimate personalized future cancer risks. We will also describe the risk-reducing intervention options included in MyLynch: colonoscopies, aspirin, weight loss, and prophylactic oophorectomies and hysterectomies. We will review the patients' responses and feedback to prototype versions of MyLynch, how those responses drove improvements and the development of new features, and describe the resulting app with screen shots and visualizations of personalized risk estimates. In the Discussion and Conclusions sections, we will review how we accomplished the four aims outlined in the previous paragraph, how our tool effectively communicates patient risks through interactive visualizations, and limitations of the tool.

Point 3: It is not clear from the paper what scientific results are included in the development web application. It would also be good to show them in the main contribution of the article. Authors should provide a link to open access repository with the dataset used for modeling

Response 3: We thank the reviewer for their thoughtful comment regarding the results. The scientific results are described in the last paragraph of the updated Introduction (see text above) as well as in the following subsections of the Results: Baseline ACP Database, Intervention Effects and the Back-end, Results from the Focus Groups and Resulting Initial Prototype, Results from the Cognitive Interviews and Iterative UI Design. The following results summary was also added to the Conclusions section:

The back-end of the app was built around a database of ACPs of gene-cancer associations, covering all five LS genes and nine different cancer types. These ACPs and the intervention effects for colonoscopies, aspirin, weight loss, oophorectomies, and hysterectomies were estimated using a rigorously curated literature review and the application of the developed statistical methods. The UI was iteratively developed based on patient feedback from 4 focus groups and 12 cognitive interviews and yielded significant improvements for the usability of the web app. 

We appreciate the reviewer’s request to include publicly accessible results. MyLynch is publicly available for free and the database is reproducible based on sources listed in table 2, and the methods described in our references: Braun, et al., 2018; NCI’s Dev Can: The probability of developing or dying from cancer, 2003-2022; and Lee, et al, 2021.

Point 4: The conclusion section should be extended using: 1) numerical results obtained in the paper; 2) limitations of the proposed solution; 3) prospects for future research.

Response 4: We agree with the reviewer that the Conclusions section needed to be expanded upon. We summarized the numerical results in the conclusion by adding the following sentences: 

The back-end of the app was built around a database of ACPs of gene-cancer associations, covering all five LS genes and nine different cancer types. These ACPs and the intervention effects for colonoscopies, aspirin, weight loss, oophorectomies, and hysterectomies were estimated using a rigorously curated literature review and the application of the developed statistical methods. The UI was iteratively developed based on patient feedback from 4 focus groups and 12 cognitive interviews and yielded significant improvements for the usability of the web app. 

We also summarized the limitations and proposed future research by adding the following two sentences to the Conclusions section:

The MyLynch risk estimates and its clinical utility should be validated on larger samples of patients in future studies. We also plan to produce a mobile compatible version of the app as well as incorporate risk uncertainty into the estimates.

Point 5: A lot of references are outdated. Please fix it using 3-5 years old papers in high-impact journals.

Response 5: We thank the reviewer for suggesting this improvement to the paper and have adjusted the sources accordingly. The following articles that are less than five years old were added to our references and are cited in the paper:

  • Dominguez-Valentin et al., 2021
  • Win et al., 2021
  • Bucksch et al., 2020

Additionally, several older references were removed and more current sources already used in the paper replaced them:

  • Watson and Lynch, 2001
  • Balas and Boren, 2000
  • Neilson, 1994 (Usability Engineering)
  • Boland and Lynch, 2013
  • Frewer, 2004

Reviewer 2 Report

Lynch syndrome (LS), formerly called hereditary non-polyposis colorectal cancer (HNPCC), affects approximately 1 out of 279 people and is associated with significantly increasedrisks, and potential earlier onsets, of a number of cancers including colorectal (CRC), endometrial, ovarian, gastric, and morePerfecting a Lynch syndrome monitoring application such as My Lynch app represents a launching pad in the modeling of therapeutic strategies, in the prevention of disease and complications, respectively in the modeling of results.

Based on using My Lynch app and consulting their clinician, the user is well informed and enabled to make lifestyle and medical intervention decisions best suited to their needs and in consultation with their clinician. Early diagnosis of LS allows medical providers to prescribe tailored screening and interventions that are highly effective in preventing cancer and, at least for colonoscopies, extending life expectancy.

The title and content of the article represent a topic of real interest worldwide. The stratification of patients with Lynch syndrome within an application represents an important point in their therapeutic management, respectively in the standardization of a target programThe subject of the study is topical with real interest for the future.

The introduction of this article presents the general advantages related to the implementation of the MyLynchapplication in order to monitor population groups at risk, respectively of patients with an established diagnosis. The bibliographic data inserted along the article presents a qualitative chronology. The subject of the article represents a true scientific revolution in its field

The material and methods section of the article presents a quantitative and qualitative exposition of the research plan, respectively a good reproducibility in order to develop other studies with this theme. I consider it necessary to develop new studies on this subject and implement them on a population scale.

The results of the article present a logical and chronological exposition outlining qualitative aspects related to the implementation of a monitoring program for the population diagnosed with Lynch syndrome, respectively the population predisposed to its development. The figures and tables keep a specific chronology throughout their exposition, presenting qualitative aspects related to the subject of the article.

The topic of the article is a real interest for the future with major importance in this field. I consider it necessary to develop new studies on this subject and implement them on a population scaleThe article presents an important research point with an optimal linguistic exposition, having an exponential potential for the future.

           This present article is written in a clear and concise manner, there by preventing or even treating this type of cancer.

           The article presents originality, with an optimal literary exposition, representing a topic of real interest for the future with objective results at the research level. The article represents a launching platform in its field and from the point of view of the characteristics it is included for publication

Author Response

Point 1: I consider it necessary to develop new studies on this subject and implement them on a population scale.

Response 1: We agree with the reviewer that MyLynch should be tested on a larger sample of patients and we have modified the text to reflect this. The following sentence in the last paragraph of the Discussion section was modified to emphasize our intent to validate MyLynch on a larger patient sample:

Additional validation studies using larger patient samples are needed to assess the accuracy of our risk estimates and to assess MyLynch's effects on cascade testing rates and its ability to influence clinical decisions.

Additionally, the following sentence was added to the Conclusions section which also emphasizes this point:

The MyLynch risk estimates and its clinical utility should be validated on larger samples of patients in future studies. 

Reviewer 3 Report

The aim of this study was to provide individualized and gene-specific cancer risks to patients with Lynch syndrome and to educate patients about relevant interventions by establishing a patient-friendly online tool. It also aimed to provide patients with adjusted risk estimates according to their chosen intervention. The concerns raised by the reviewer are listed below:

1.  In the introduction, please provide examples of so-called pathogenic variants in the four mismatch repair genes.

2.  In the introduction, please introduce the available methods used to detect pathogenic variants in genes associated with Lynch syndrome.

3. If a patient has a random pathogenic variant in two independent mismatch repair genes, how does that patient acquire an accurate cancer risk? 

4. Similar applications may be explored for Peutz-Jeghers syndrome in the future. This can be discussed in the discussion section.

Author Response

Point 1: In the introduction, please provide examples of so-called pathogenic variants in the four mismatch repair genes.

Response 1: We agree with the reviewer’s comment and think that addressing it adds value to the Introduction section and readers’ understanding of LS. The following sentence was modified in the Introduction to provide examples of types of pathogenic variants in MMR genes:

[LS] is a hereditary disease with an autosomal dominant pattern caused by a pathogenic variant (PV) (primarily missense, nonsense, frameshift, and splice site change variants) in one of four mismatch repair (MMR) genes: MLH1, MSH2, MSH6, and PMS2, or a PV with a large deletion in the 3' region of the EPCAM gene which, in turn, causes epigenetic silencing of MSH2.

Additionally, the following sentence was added to the Introduction to demonstrate how risk varies based on which gene harbors a PV:

According to Dominguez-Valentin (2020), a female LS patient with a MLH1 PV is estimated to have a 37% cumulative risk of endometrial cancer by age 75 whereas if they instead carried an EPCAM PV then their risk would only be around 13%.

Point 2: In the introduction, please introduce the available methods used to detect pathogenic variants in genes associated with Lynch syndrome.

Response 2: We agree with the reviewer’s comment. The available methods to detect pathogenic variants in LS genes are described in the following sentence and were added to the Introduction section based on Phillips et al, 2018:

A patient's blood or saliva sample can be used to identify the presence of germline PVs on single genes, multiple gene panels, exomes, or whole genomes using microarrays, sequencing, and karyotyping.

Point 3: If a patient has a random pathogenic variant in two independent mismatch repair genes, how does that patient acquire an accurate cancer risk? 

Response 3: The reviewer's comment brings up a unique situation that deserves discussion within the paper. Due to the  limited amount of available literature on multiple PVs on different MMR genes, likely due to the rarity of such a condition, assessing cancer risk for these cases was out of the scope of this work. We modified the following sentence in the Discussion section to make it clear that this is a limitation:

There is also limited literature available on cancer risks for individuals with PVs in two different LS genes therefore, MyLynch does not provide tailored risk estimates specifically for these individuals. However, we recommend that these patients compare the future risk estimates, by cancer type, for each of their PV carrying genes and that they assume the highest risk conferring gene among the two provides the best estimation of their personalized cancer risk. 

Point 4: Similar applications may be explored for Peutz-Jeghers syndrome in the future. This can be discussed in the discussion section.

Response 4: We think that the reviewer brings up a good point on how this work can be expanded upon to other inherited cancer syndromes. The following sentence about extending our methods for other hereditary cancer syndromes was added to the Discussion section in the paragraph that talks about future steps:

The methods used to create this web app can also be extended to benefit patients of other, non-LS, inherited cancer syndromes such as hereditary breast and ovarian cancer (HBOC), Peutz-Jeghers syndrome, and more.

Round 2

Reviewer 1 Report

Paper can be accepted in current form